# Loss of SET1/COMPASS methyltransferase activity reduces lifespan and fertility in *Caenorhabditis elegans*

Matthieu Caron[1], Loïc Gely[1], Steven Garvis[1], Annie Adrait[2], Yohann Couté[2], Francesca Palladino[1], Paola Fabrizio[1]

Changes in histone post-translational modifications are associated with aging through poorly defined mechanisms. Histone 3 lysine 4 (H3K4) methylation at promoters is deposited by SET1 family methyltransferases acting within conserved multiprotein complexes known as COMPASS. Previous work yielded conflicting results about the requirement for H3K4 methylation during aging. Here, we reassessed the role of SET1/COMPASS–dependent H3K4 methylation in *Caenorhabditis elegans* lifespan and fertility by generating *set-2(syb2085)* mutant animals that express a catalytically inactive form of SET-2, the *C. elegans* SET1 homolog. We show that *set-2(syb2085)* animals retain the ability to form COMPASS, but have a marked global loss of H3K4 di- and tri-methylation (H3K4me2/3). Reduced H3K4 methylation was accompanied by loss of fertility, as expected; however, in contrast to earlier studies, *set-2(syb2085)* mutants displayed a significantly shortened, not extended, lifespan and had normal intestinal fat stores. Other commonly used *set-2* mutants were also short-lived, as was a *cfp-1* mutant that lacks the SET1/COMPASS chromatin-targeting component. These results challenge previously held views and establish that WT H3K4me2/3 levels are essential for normal lifespan in *C. elegans*.

## Introduction

Epigenetic alterations, such as post-translational histone modification and DNA methylation, are well-established hallmarks of cellular senescence and organismal aging (1, 2). Two important consequences of these modifications are transcriptional deregulation and increased genome instability, which are thought to play key roles in aging and age-related diseases (1). However, the complexity of the aging epigenome, especially in mammals, has hampered efforts to establish whether and how individual epigenetic alterations contribute to the aging process. Studies in simple model organisms have provided valuable insight into this question. Work in yeast, worms, and flies has shown that nutrient-regulated

signaling pathways, such as the insulin/IGF-1 and target of rapamycin pathways, play conserved roles in the regulation of mammalian lifespan (3, 4). More recent studies in these model systems have sought to establish a causal link between specific epigenetic modifications and the aging process. For example, overexpression of H3 and H4 histones in WT budding yeast is sufficient to extend lifespan, suggesting that the observed loss of histones during replicative aging directly contributes to the aging process (5, 6). Global loss of histones is also observed in aging *Caenorhabditis elegans*, senescent human cells, and human fibroblasts undergoing replicative aging (7, 8, 9), suggesting that a decline in core histone proteins may represent a conserved aging mechanism.

Loss of heterochromatin is another feature widely regarded as a contributor to the aging process (10). In aging organisms, senescent cells, and models of progeroid syndromes, loss of heterochromatin is associated with a decrease in the abundance of the repressive trimethylated histone 3 lysine 9 (H3K9me3) mark and of heterochromatin protein 1 (HP1) (10, 11). The abundance of another repressive mark, H3K27me3, also declines with time, but the manner in which aging affects this mark differs substantially not only between organisms but also between tissues within the same organism (2, 11). In *C. elegans*, loss of H3K27me3 in aging somatic cells is accompanied by transcriptional up-regulation of UTX-1, a demethylase responsible for H3K27me3 removal (12). Notably, reducing UTX-1 levels by RNAi reverses the age-dependent decline in H3K27me3 and prolongs lifespan (12, 13), suggesting that maintenance of this mark is associated with a delay in chronological aging. In contrast, *Drosophila* carrying mutations that affect polycomb repressor complex 2 components have reduced global levels of H3K27me3, but the males exhibit an extended lifespan (14). Although the reason for these opposing effects of H3K27me3 removal on the lifespans of worms and flies is unclear, it most likely involves altered expression of a few key target genes rather than global changes in H3K27me3 distribution (13). Similarly, misregulation of distinct sets of target genes may also account for the extended lifespan of *C. elegans* induced by either inactivation or overexpression of UTX-1 (15).

Histone modifications associated with active chromatin, such as H3K4me3 and H3K36me3, are also subject to age-dependent

---

[1]Laboratory of Biology and Modelling of the Cell, Ecole Normale Supérieure de Lyon, CNRS UMR5239, INSERM U1210, Université de Lyon, Lyon, France   [2]University of Grenoble Alpes, INSERM, CEA, UMR BioSanté U1292, CNRS, CEA, FR2048, Grenoble, France

Correspondence: francesca.palladino@ens-lyon.fr; paola.fabrizio@ens-lyon.fr

deregulation and have been implicated in lifespan regulation (16). H3K4me3, a mark associated with active promoters, has context-dependent functions in gene regulation (17). In yeast, H3K4 methylation is deposited exclusively by the methyltransferase Set1, which acts as part of the multisubunit complex of proteins associated with Set1 (COMPASS) (18, 19). Mammals express at least six proteins capable of methylating H3K4: two Set1-related proteins, Set1A and Set1B, and four more distantly related MLL proteins (MLL-1–4), which are found in distinct SET or MLL COMPASS complexes that share the core regulatory components Swd3/WDR5, Swd1/RbBP5, Bre2/ASH2, and Sdc1/hDPY30 (17, 20, 21, 22, 23, 24, 25, 26). In the mouse, all six Set1 and MLL proteins have been shown to be essential for survival (17). Whereas SET1/COMPASS complexes control global H3K4 methylation, MLL/COMPASS–like complexes methylate H3K4 at specific gene targets or regulatory regions (17). In *C. elegans*, the single homologs of SET1 and MLL, SET-2 and SET-16, respectively (27, 28), also display different functions. SET-2 deposits most H3K4me3 marks in both the germline and soma (27, 28), and worms lacking SET-2 are viable but have reduced fertility; in contrast, loss of SET-16 causes embryonic lethality, consistent with studies of mammalian MLL proteins (29, 30).

The role of H3K4me3 in lifespan regulation is controversial and conflicting results have been obtained with several model organisms. Yeast *set1* mutants are short-lived in both replicative and chronological aging paradigms (31, 32, 33). Set1-dependent H3K4me3 promotes chronological longevity in yeast by preventing apoptosis and histone loss, and the mark also contributes to the maintenance of transcriptional patterns required for a normal replicative lifespan (31, 32, 33). Conversely, partial loss-of-function mutations or knockdown of genes encoding the SET1/COMPASS components SET-2, WDR-5.1, and ASH-2 in *C. elegans* have been shown to extend lifespan through changes in lipid metabolism triggered by the germline (34, 35). The lifespan-extending effect observed in *wdr-5.1* and *set-2* mutants was transmitted trans-generationally, and WT descendants of *wdr-5.1* or *set-2* mutants over several generations were also long-lived (36). More recently, this transgenerational effect was attributed to a generation-dependent accumulation of repressive H3K9me2 (37). However, these studies did not investigate whether loss of H3K4 methylation or acquisition of H3K9 methylation may have contributed to the effects by directly altering the expression of germline genes that affect longevity.

To address the role of H3K4 methylation in *C. elegans* lifespan, we generated a mutant that is unable to mediate H3K4 methylation but retains the ability to form a SET1/COMPASS complex, thereby avoiding interference with potential non-catalytic complex activities. To this end, we generated mutants carrying a catalytically inactive allele of *set-2* using CRISPR/Cas9 editing and analyzed the effects on SET1/COMPASS complex assembly, H3K4 di- and trimethylation, fat accumulation, fertility, and, most importantly, lifespan. Interestingly, mutants with catalytically inactive SET-2 displayed some, but not all, of the previously characterized phenotypes of *C. elegans* mutants harboring varying degrees of SET-2 loss of function, with the most striking difference being the shortened, rather than extended, lifespan of our mutants compared with WT animals (34). Overall, the results of this study suggest a more complex, and likely context-dependent, requirement

for SET1/COMPASS-mediated H3K4 methylation in the maintenance of normal lifespan in *C. elegans*.

# Results

## A catalytically inactive *set-2* allele reduces global H3K4 di- and trimethylation in *C. elegans*

Earlier studies suggesting that loss of H3K4 trimethylation extends *C. elegans* lifespan were performed using *set-2* RNAi knockdown or a *set-2* partial loss-of-function allele, *set-2(ok952)* (34, 35, 36). The *set-2(ok952)* mutation is a complex insertion/deletion that encodes a SET-2 protein with an intact catalytic SET domain but missing an upstream region (Fig 1A); accordingly, *set-2(ok952)* animals show only a small reduction in global H3K4me3 levels in the soma and germline and no obvious phenotype (27). To more directly investigate how SET1/COMPASS–dependent methylation contributes to lifespan in *C. elegans*, we engineered a catalytically inactive *set-2* allele, *set-2(syb2085)*, by replacing the highly conserved histidine (H) residue at amino acid position 1447 of the SET domain with lysine (K) using CRISPR-Cas9 technology (Fig 1A). This residue is conserved in the great majority of SET domain proteins regardless of the histone substrate specificity, and structural studies as well as in vitro and in vivo assays have demonstrated its requirement for histone methyltransferase activity (38, 39, 40, 41, 42, 43, 44, 45, 46, 47). Immunofluorescence staining of *set-2(syb2085)* animals revealed a marked decrease in H3K4me3 (~80%) in the germline compared with WT (N2) animals (Fig 1B and C). Moreover, the H3K4me3 reduction in *set-2(syb2085)* animals was comparable to that observed in *set-2(bn129)* null mutants previously described (29) and in *cfp-1(tm6369)* mutants lacking the SET1/COMPASS component CFP-1 (48) (Fig 1B and C). Consistent with the earlier work, H3K4me3 levels in the germline of *set-2(ok952)* animals were relatively modestly reduced compared with WT worms (27) (Fig 1B and C). Similar results were obtained when the germlines were immunostained for H3K4me2, which is also mediated by SET1/COMPASS in *C. elegans* (Fig 1B and C) (29). The immunostaining results were confirmed by Western blot analysis of mixed staged embryos followed by semi-quantitation of H3K4me2 and H3K4me3 (Fig 2A and B). No significant differences in H3K4me2/3 levels were detected between WT and *set-2(ok952)* embryos, whereas both histone marks were reduced by ~60% in *set-2(bn129)*, *set-2(syb2085)*, and *cfp-1(tm6369)* mutants. Taken together, our results demonstrate that the levels of H3K4me2/3 in *set-2(syb2085)* animals are reduced to levels comparable to those seen in *set-2* null animals, consistent with complete loss of *set-2* catalytic activity. The residual H3K4me2/3 observed in both germlines and embryos is most likely due to SET-16, the only other known SET1 family protein in *C. elegans* (29, 49).

To verify that loss of SET-2 catalytic activity in *set-2(syb2085)* mutants does not interfere with SET-2 protein stability or SET1/COMPASS integrity, we performed co-immunoprecipitation and mass spectrometry–based proteomic analyses using a newly generated *C. elegans* strain expressing a *gfp::cfp-1* knock-in allele [*cfp-1(syb1012)*]. These experiments showed that immunoprecipitates of GFP::CFP-1 from lysates of *set-2(syb2085)* mutant embryos

Figure 1. **Loss of SET-2 enzymatic activity in** *Caenorhabditis elegans* **markedly reduces H3K4me3 and H3K4me2 in the germline.**
**(A)** Upper panel: schematic showing the *set-2* alleles used in this study. Black and shaded boxes represent conserved domains (RRM, SET, and post-SET). The position of the amino acid substitution introduced within the SET domain of *set-2(syb2085)* is shown. Asterisk indicates a 12 bp-insertion present in the *set-2(ok952)* allele. Lower panel: sequence alignment of the conserved SET domain motif (black box) from selected H3K4 histone methyltransferases including the His residue mutated in *set-2(syb2085)* (red box). **(B)** Schematic showing the regions of the germline used for quantification of immunofluorescence signals: mitotic zone (MZ), transition zone (TZ), and pachytene (PA). DTC, distal tip cell. **(C)** Representative confocal images of dissected gonads from WT animals and the indicated *set-2* or *cfp-1* mutants stained with Hoechst and anti-H3K4me3 antibody (left panels) or anti-H3K4me2 (right panels). Scale bar, 10 μm. Lower bar graphs show quantification of H3K4me3/2 signals normalized to Hoechst in the MZ + TZ and PA regions. Data show changes in the mean pixel intensities expressed as percent of the WT ± SD, n = 10. ns, not significant, **$P < 0.005$, ****$P < 0.0001$ for mutants versus WT worms by one-way ANOVA followed by Tukey's test.

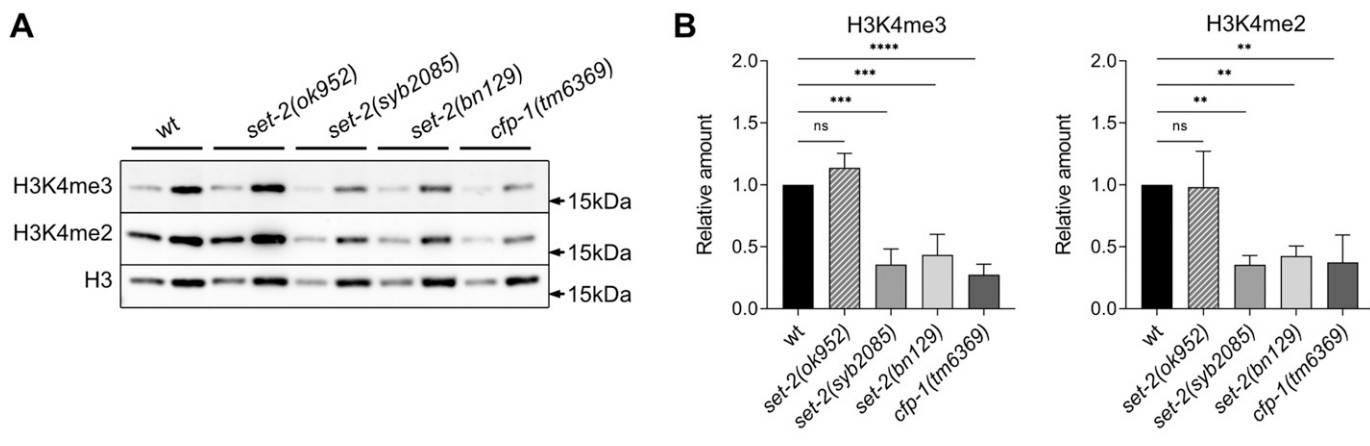

**Figure 2. Loss of SET-2 catalytic activity causes a global reduction in H3K4me3 and H3K4me2 in *Caenorhabditis elegans* embryo.**
**(A)** Western blot analysis of proteins from mixed stage embryos of WT and the indicated *set-2* or *cfp-1* mutants. Blots were probed with anti-H3K4me3, anti-H3K4me2, and anti-H3 antibodies. Left and right lanes for each strain represent samples with 1 and 3 μg total protein, respectively. **(A, B)** Quantification of H3K4me3 (left) and H3K4me2 (right) band intensities from blots represented in (A). Mean ± SD, n = 3. ns, not significant, **P < 0.01, ***P < 0.001, ****P < 0.0001 for mutants versus WT worms by one-way ANOVA followed by Dunnett's test. **(C)** List of selected proteins (and their mammalian homologs) identified by mass spectrometry of proteins immunoprecipitated with GFP::CFP-1 from embryos carrying the indicated *set-2* alleles. SET-2/SET1 complex specific components and SET1/MLL core complex components are highlighted in light and dark gray, respectively. SC, spectral counts. Control samples were obtained from WT animals not carrying the GFP::CFP-1 allele.

contained SET-2, the core SET1/COMPASS components ASH-2, RBBP-5, WDR-5.1, and DPY-30, and the unique subunit SWD-2.1 (Fig 2C and Table S1). Similar results were obtained in CFP-1 immunoprecipitates from *set-2(ok952)* mutant embryos (Fig 2C and Table S1), whereas none of the SET1/COMPASS subunits were detected in immunoprecipitates from *set-2(bn129)* null mutants (Table S2). Taken together, these results indicate that SET1/COMPASS formation is maintained in *set-2(syb2085)* mutant animals, and indicate that disruption of the complex is not responsible for the loss of H3K4 methylation in these mutants.

## Partial or complete *set-2* inactivation or loss of SET-2 catalytic activity shortens lifespan

As noted above, previous work suggested that loss of H3K4 trimethylation in *set-2(ok952)* mutants results in lifespan extension, potentially via modulation of fat metabolism (35). To determine whether loss of H3K4me2/3 in the catalytically inactive SET-2 mutants was associated with similar effects on lifespan, we compared *set-2(syb2085)* animals with N2 WT worms and *set-2(ok952)*, *set-2(bn129)*, and *cfp-1(tm6369)* mutants, all of which display varying H3K4me2/3 levels. Unexpectedly, we observed a significant and reproducible reduction in the mean lifespan not

only of *set-2(syb2085)*, *set-2(bn129)*, and *cfp-1(tm6369)* mutants but also of *set-2(ok952)* mutants, which had previously been described as exhibiting increased longevity compared with WT animals (Fig 3A and Table S3). Our finding that each of the SET1/COMPASS mutants displayed decreased lifespans compared with WT animals are consistent with recent work showing similar effects on longevity in *set-2(bn129)* mutants, *ash-2* mutants (50), and early generations of *wdr-5* mutants (37).

Because the previous study examining *set-2(ok952)* animals attributed the extended lifespan to altered lipid metabolism (35), we used Oil Red O staining to quantify levels of neutral lipids, the main form of lipid storage in *C. elegans*. However, we observed no significant differences in neutral lipid content between WT animals and either *set-2(ok952)* or *set-2(syb2085)* animals. As previously shown, lipid stores were increased in insulin signaling *daf-2(e1370)* mutants that show extended lifespan, and decreased in *eat-2(ad1116)* mutants defective in pharyngeal pumping (Fig 3B and C and Table S3) (51, 52, 53). Consistent with our results, in an independent study lipid content was found to be unaltered in *wdr-5* loss-of-function mutants (54). These results demonstrate that loss of catalytic activity in SET1/COMPASS and a corresponding reduction in global H3K4me2/3 not only shortens lifespan in *C. elegans* but also does so without modifying neutral lipid content.

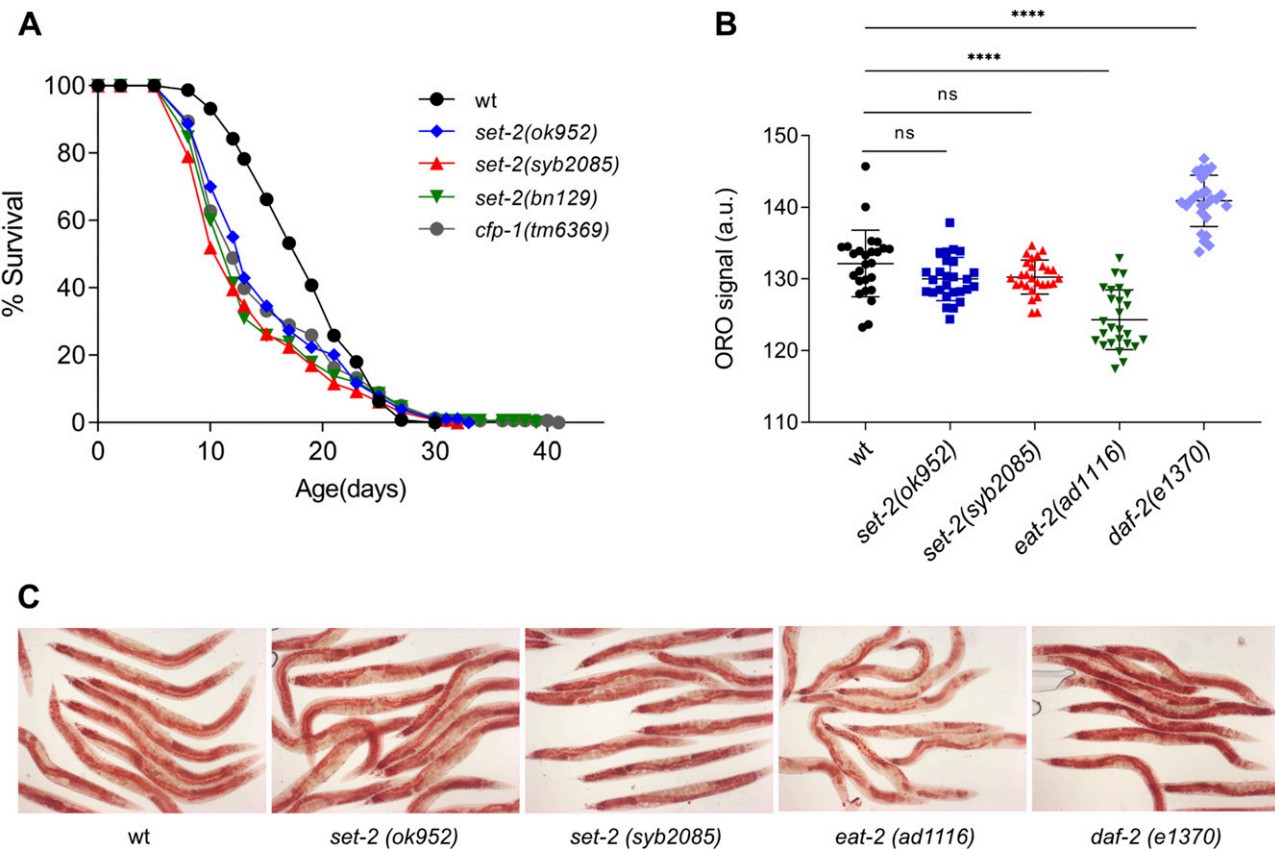

**Figure 3. Loss of SET-2-dependent methylation reduces lifespan.**
**(A)** Lifespan assays of WT *Caenorhabditis elegans* and *set-2* and *cfp-1* mutants. Mean reductions relative to WT animals are 17.6% for *set-2(ok952)* (*P* < 0.001), 26% for *set-2(syb2085)* (*P* < 0.0001), 23.3% for *set-2(bn129)* (*P* < 0.0001), and 18.3% for *cfp-1(tm6369)* (*P* < 0.001). *P*-values were calculated using the Mantel–Cox log-rank method. Replicate experiments are shown in Table S3. **(B, C)** Quantification of Oil Red O signal intensity and (C) representative images of Oil Red O–stained WT, *set-2(ok952)*, *set-2(syb2085)*, *eat-2(ad1116)*, and *daf-2(e1370)* animals. Magnification = 10×. Data represent the mean intensity ± SD of n = 27 worms. ns, not significant; ****P < 0.0001 by one-way ANOVA followed by Tukey's test.

### Loss of SET-2–dependent H3K4 methylation is responsible for loss of fertility

We considered that the reduced lifespan exhibited by *set-2* mutants could result from developmental defects (55). However, we found that the development time of *set-2(syb2085)* and *set-2(ok952)* animals was only modestly lengthened compared with that of WT animals (~67, ~64, and ~62 h, respectively; Fig S1). This finding suggests that the shortened lifespan of *set-2(syb2085)* animals is unlikely to be due to major developmental defects, although we cannot exclude a contribution from more subtle defects in neuronal development (56).

Inactivation of SET1/COMPASS components has been shown to disrupt (29) or have no effect (34) on *C. elegans* fertility. Therefore, we analyzed fertility phenotypes in *set-2(syb2085)*, *set-2(bn129)*, and *set-2(ok952)* animals. We found that loss of SET-2 catalytic activity resulted in fertility defects similar to those of *set-2* null animals, including decreased brood size under normal growth conditions (20°C) and progressive sterility at a higher temperature (25°C), although the defects were less severe than in *set-2(bn129)* mutants (Fig 4A and B) (57). As previously reported, the fertility phenotypes of *set-2(ok952)* animals were not significantly different from those of WT animals (Fig 4A and B) (57). Collectively, these results indicate that

SET1/COMPASS-dependent H3K4 di- and trimethylation play a crucial role in maintaining germline homeostasis.

## Discussion

Previous studies using partial knockdown of SET1/COMPASS subunits have led to the commonly held view that loss of H3K4 trimethylation increases lifespan in *C. elegans* (1, 2). More recent studies suggested that this effect is observed only after multiple generations of knockdown of SET1/COMPASS components and depends on the accumulation of repressive H3K9me2 (37). In the present work, we sought to clarify whether SET1/COMPASS–dependent H3K4 methylation affects longevity by examining *C. elegans* mutants in which the catalytic activity of SET-2 was selectively inactivated, thereby leaving other functions of the protein and/or COMPASS unaltered. We found not only that SET-2-mediated H3K4 methylation is essential for a normal lifespan, but also that, in contrast to earlier studies, even a mild reduction in SET-2 activity is detrimental to lifespan, as demonstrated by *set-2(ok952)* animals.

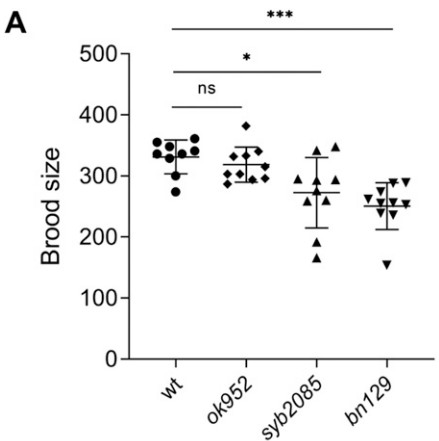

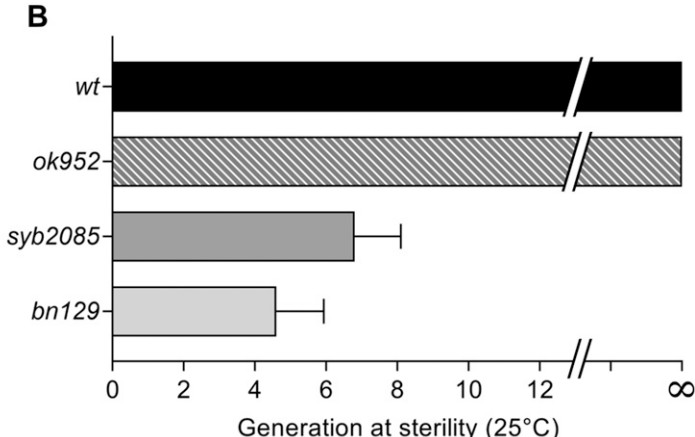

**Figure 4. SET-2 catalytic activity is required for germline immortality.**
**(A)** Total number of progeny from WT, *set-2(ok952)*, *set-2(syb2085)*, and *set-2(bn129)* animals grown at 20°C. Data represent the mean ± SD, n = 10. ns, not significant; *P < 0.05, ***P < 0.001 for *set-2(syb2085)* and *set-2(bn129)* versus WT, respectively, by one-way ANOVA followed by Tukey's test. **(B)** Transgenerational fertility assay of WT animals and the indicated *set-2* mutants performed at 25°C. Scoring was based on five biological replicates, with six independent lines each, error bars show SD. WT animals can be maintained for more than 40 generations without loss of fertility (48).

We observed decreased longevity in *set-2(bn129)* mutants, which as shown here fail to form a SET1/COMPASS complex, as well as in *set-2(syb2085)* animals. Both mutants also exhibited similar global loss of H3K4me2/3 methylation in the germline and soma, decreased fertility at 20°C, and progressive sterility at the stressful temperature of 25°C. Recent reports for several histone-modifying complexes suggest that they play important functions independent of catalytic activity (58). Our results strongly suggest that the shortened lifespan and loss of fertility phenotypes of *set-2* mutants result from loss of H3K4 di- and trimethylation rather than catalytic-independent functions of SET-2. Of note, we found that *cfp-1* mutants lacking the CFP-1 chromatin-targeting subunit of COMPASS also exhibited reduced lifespan. Thus, although we cannot exclude that loss of methylation of a non-histone substrate contributes to the observed phenotypes of *set-2(syb2085)* animals, our results are most consistent with a crucial role for SET1/COMPASS–dependent H3K4 methylation in maintaining a normal lifespan, in addition to its previously documented role in promoting germ cell survival (29, 49, 57).

An earlier study reported that the lifespan of *set-2(ok952)* mutant animals was ~20% longer than that of WT animals (34), which contrasts with observations in the present study. We were also unable to recapitulate the increase in lipid storage previously implicated in the lifespan extension of *set-2(ok952)* animals (35). Indeed, neither *set-2(ok952)* nor *set-2(syb2085)* mutants showed a detectable change in intestinal lipid content compared with WT worms in our study. We found that the mutant SET-2 protein in *set-2(ok952)* animals retains its ability to associate with other SET1/COMPASS components and exhibits only a small decrease in H3K4 methylation compared with WT animals, consistent with the presence of an intact SET-2 catalytic domain (27, 49). The causes for the discrepancies in lifespan of *set-2(ok952)* animals in the present and earlier study could be partly related to subtle experimental differences such as reagent purity and quality, or small variations in incubator temperature and/or humidity (59, 60). The genetic background of *C. elegans* mutants has been shown to influence

lifespan (61), and additional mutations could have contributed to the observed differences in the lifespan of *set-2(ok952)* mutants (34). More recent studies suggest that changes in lifespan may not be observed for ≥20 generations in animals with reduced H3K4 methylation, as shown for *C. elegans* mutants lacking WDR-5.1 (37). In the present study we did not test whether the detrimental effect on lifespan caused by loss of SET1/COMPASS activity is reversed after multiple generations. We note, however, that not only is WDR-5.1 a common component of both SET1/ and MLL/COMPASS complexes, but is also present in other multiprotein complexes with histone-modifying activity in *C. elegans* and other species (48, 62, 63, 64, 65). In addition, WDR-5.1 also has established COMPASS/H3K4 methylation-independent functions (66). Therefore, the context in which WDR-5.1 influences lifespan remains to be established and a SET1/COMPASS–independent mechanism is plausible.

Our results show a direct correlation between loss of H3K4me2/3 in the germline and fertility defects in *C. elegans*. Although *set-2(syb2085)* and *set-2(bn129)* mutants became rapidly sterile at 25°C, *set-2(ok952)* mutant animals remained fertile, consistent with only a modest loss of H3K4me2/3 in these animals. Sterility in mutants lacking SET-2 catalytic activity may be related to altered chromatin organization or global deregulation of the germline transcriptional program, as we previously observed in the *set-2(bn129)* mutants (67, 68). Interestingly, a similar correlation was not observed between H3K4 methylation and the longevity phenotype, in that all three *set-2* mutants showed a reduction in lifespan despite having markedly different global H3K4me2/3 levels. These results suggest that selective alteration in methylation patterns on specific genes, rather than a global decrease, may be responsible for the observed phenotype. In this regard, changes in H3K27me3 levels that affect the expression of relatively few genes have been shown to influence *C. elegans* lifespan (12, 13). Likewise, SET1/COMPASS–dependent H3K4me3 maintains normal chronological lifespan in yeast by specifically promoting histone gene expression (32). Our results could also be explained by the existence of a critical threshold of global H3K4me2/3 below which longevity is decreased.

The results of our study provide novel insights into the controversy surrounding the impact of H3K4 methylation on longevity. In yeast, loss of SET1/COMPASS significantly shortens replicative lifespan, most likely via reduced expression of a set of genes that are normally induced during aging (33). In *C. elegans*, over-expression of the RBR-2 H3K4 demethylase extends lifespan (34), whereas mutations or RNAi-mediated knockdown of the same enzyme have been shown to either increase (7, 69, 70) or decrease (12, 34) lifespan, depending on the allele used. Decreased longevity in *rbr-2* mutants has been attributed to the presence of an additional mutation in the original genetic background (69). In *Drosophila melanogaster*, RNAi-mediated depletion or mutation of Lid, the RBR-2 ortholog, resulted in increased levels of H3K4me3 and reduced lifespan in males (71). Interestingly, depletion of LSD-1, another H3K4 demethylase targeting H3K4me1 and H3K4me2, has been shown to extend lifespan in *C. elegans* (12, 72). Taken together, these data and the results of the present study argue against a simple model in which H3K4 methylation and demethylation play opposing roles in lifespan regulation (34). Additional studies using well-characterized mutants and tissue-specific RNAi/deletion approaches will be required to clarify this complex relationship and to establish the contexts in which selective loss of SET1/COMPASS-dependent H3K4 methylation alters the chromatin landscape during aging.

# Materials and Methods

### Genetics and strains

Strains were cultured under standard laboratory conditions (73). The WT N2 (Bristol) was used as reference strain. The strains used are as follows (name, genotype, and origin): PFR401, *set-2(ok952)* III, CGC; PFR253, *set-2(bn129)* III, (29); PFR588, *cfp-1(tm6369)* IV, NBRP; PHX2171, *set-2(syb2085)*/qC1[*dpy-19(e1259) glp-1(q339)* qIs26] III, this work; PHX1012, *cfp-1(syb1012)* IV, this work; PFR727, *cfp-1(syb1012)* IV; *set-2(ok952)* III, this work; PFR728, *cfp-1(syb1012)* IV; *set-2(syb2085)* III, this work; PFR558, qaIs22[*HA::wdr-5.1;Cbunc-119(+)]*; *set-2(bn129)* III; *wdr-5(ok1417)* III (48); CB1370, *daf-2(e1370)* III, CGC; XA6226, *mrg-1(qa6200)*/qC1 [*dpy-19(e1259) glp-1(q339) qIs26*] III, CGC; and DA1116, *eat-2(ad1116)* II, CGC.

All mutants were outcrossed to WT N2 at least four times. *cfp-1(syb1012)* and *set-2(syb2085)* were generated by CRISPR/Cas9 technology at SunyBiotech (Fuzhough City, Fujian, China). *cfp-1(syb1012)* is a knock-in allele carrying a degron-GFP tag inserted at the 5' end of the endogenous *cfp-1* coding sequence. For *set-2(syb2085)*, the highly conserved histidine residue at amino acid position 1447 within the catalytic SET domain was replaced by lysine (38, 39). Two synonymous point mutations were introduced to avoid cutting of the repair template by Cas9. WT and *syb2085* sequences are shown below, with the H1447K substitution in bold and synonymous mutations capitalized:

wt: aagcgtggaaattttgctcgatttatt AAT **CAC** tcgtgc CAA cctaattgctacgcgaaggta
*syb2085*: aagcgtggaaattttgctcgatttatt AAC **AAG** tcgtgc CAG cctaattgctacgcgaaggta

To avoid the potential accumulation of epigenetic modifications (37), *set-2(syb2085)* mutants were maintained in a heterozygous state by crossing them with XA6226 carrying the *qC1* balancer chromosome (74).

### Lifespan analysis

Lifespan assays were conducted at 20°C according to standard protocols (75). Worms were thawed and grown for three to four generations at 20°C. Worms were synchronized by bleaching and transferred at the L1 stage. Worms were maintained on solid Nematode Growth Medium (NGM) seeded with *Escherichia coli* strain OP50-1. Solid NGM was prepared by mixing 3 g of NaCl (Euromedex), 2.5 g of peptone (BD), and 20 g of agar (Euromedex) in 1 liter of distilled water. After autoclaving 1 ml of cholesterol (Sigma-Aldrich [5 mg/ml in ethanol]), 1 ml of 1 M $MgSO_4$ (Sigma-Aldrich), 1 ml of 1 M $CaCl_2$ (Euromedex), and 25 ml of 1 M (pH 6.0) $KPO_4$ (Euromedex) were added. The medium was poured onto 35-mm vented plates. 5-Fluoro-2'-deoxyuridine was not added to the plates. The L4 stage was considered day 0. Hermaphrodites were transferred onto fresh seeded plates every day for the first week of the lifespan assay. Animals that failed to display heat-provoked movement were scored as dead, and animals that crawled off the plates or "bagged" worms were censored. *P*-values were calculated using the log-rank (Mantel–Cox) method. Statistical analysis of individual lifespan studies was performed using the Oasis online software (76). Statistical analysis of experiments shown in the main text and replicate experiments are provided in Table S3.

For experiments using the balanced strain PHX2171, non-roller *set-2(syb2085)* homozygotes were isolated and grown at 20°C for at least two generations before initiating the analysis to avoid the maternal effect.

### Brood size assay

Ten L4 larvae were isolated on individual plates seeded with OP50-1 bacteria at 20°C and allowed to develop into egg-laying adults overnight. Adult animals were then transferred to fresh plates every 12 h until reproduction ceased. Plates were scored for the number of viable progeny 24 h after removal of the mother.

### Developmental time measurement

Twenty gravid adults were allowed to lay eggs on NGM plates spotted with OP50-1 for 1 h and then removed. The time to egg-laying adulthood was measured from this point. Eggs were allowed to hatch and 10 animals were randomly picked and transferred to individual plates. Plates bearing single young adults were checked every hour for the presence of eggs.

### Transgenerational fertility assays

Six independent lines were established from freshly thawed WT animals, *set-2(ok952)* and *set-2(bn129)* mutants, and homozygous *set-2(syb2085)* mutants obtained from balanced strain PHX2171. For each line, six homozygous L4 stage animals were transferred to single plates seeded with OP50-1 and grown at 25°C. From each

generation, six worms were again picked and transferred to single plates until the animals became sterile (fewer than 10 larvae/plate), as previously described (29). When at least 200 larvae were counted on a plate, the line was considered fully fertile.

## Oil Red O staining

Worms were synchronized by bleaching and grown to the young adult stage at 20°C on NGM plates seeded with OP50-1. Worms were washed off the plates with 1× phosphate-buffered saline containing 0.1% Tween (PBST), placed in Eppendorf tubes, pelleted at 150$g$ for 30 s, and washed three times with PBST. Worms were incubated with rocking for 3 min in 600 $\mu$l of 40% isopropanol, the isopropanol was then removed, and the worms were mixed with 600 $\mu$l of Oil Red O (3:2 ratio of 5 mg/ml Oil Red O/100% isopropanol and sterile water). Animals were incubated on a rotator at RT for 2 h, pelleted at 150$g$, and washed for 30 min in 600 $\mu$l PBST. Aliquots of 10 $\mu$l of stained animals were deposited onto freshly prepared 3% agarose pads, covered with coverslips, and imaged with the same exposure settings at 10× magnification with a Zeiss AxioImager Z1 equipped with a CoolSNAP color camera. Quantification of staining was performed using FIJI image processing software (35). Briefly, the original TIFF images were background subtracted, converted to grayscale, inverted, and thresholded. The intestinal cells of whole worms were selected using the oval brush tool, saved to the Region of Interest manager, and mean Oil Red O intensity was quantified. Experiments were performed twice with similar results.

## Western blot analysis

Gravid animal were grown on NGM plates at 20°C and harvested for hypochlorite treatment. Mixed staged embryos (<100 cells) obtained by this procedure were collected, pelleted in M9 buffer and flash frozen in liquid nitrogen. Pellets were recovered in TNET buffer (50 mM Tris–HCl pH 8.0, 300 mM NaCl, 1 mM EDTA, and 0.5% Triton X-100 supplemented with cOmplete Easypack protease inhibitor cocktail [Sigma-Aldrich]). Samples were mixed 1:1 by volume with Lysing Matrix Y (MP Biomedicals) and lysed using a Precellys 24 homogenizer (Bertin) at two bursts of 20 s ON at 6,000 rpm with 10 s OFF between bursts. Samples were then centrifuged at 4°C for 5 min at 20,000$g$ and protein concentrations in supernatants were measured using the Bradford assay (Bio-Rad Protein Assay Dye). Aliquots equivalent to 1 and 3 $\mu$g protein were separated on NuAGE 12% Bis-Tris Gels using MOPS 1X (Invitrogen) running buffer, and then transferred to 0.45 $\mu$M nitrocellulose membranes (Bio-Rad Laboratories). Membranes were blocked with PBST containing 5% non-fat dry milk, and incubated overnight at 4°C with anti-H3 (#14269 at 1:2,500 dilution in PBST containing 1% BSA; Cell Signaling Technology), anti-H3K4me2 (#39913 at 1:2,000; Active Motif), or anti-H3K4me3 (#A2357 at 1:2,000; AB Clonal) primary antibodies. The membranes were washed then four times in PBST, incubated with secondary antibodies: IRDye 800CW goat anti-Rabbit (# 926-32211 at 1:10,000; LI-COR Biosciences) and IRDye 680 RD goat anti-Mouse (# 925-68070 at 1:10,000; LI-COR Biosciences). Images were obtained with the Chemidoc MP Imaging System (Bio-Rad Laboratories) and

fluorescent signals were quantified using Image Lab software (Bio-Rad Laboratories).

## Immunofluorescence microscopy

Gonads were fixed essentially as previously described (77) and incubated overnight at 4°C with mouse anti-H4K3me3 (#305-34819 at 1:500 dilution in PBST; Wako) or rabbit anti-H3K4me2 (#710796 at 1:20,000; Invitrogen) primary antibodies. Slides were washed with PBST, and incubated at RT for 50 min with goat anti-mouse Alexa Fluor Plus 555 (#A32727 at 1:1,000; Invitrogen/Probes) or goat anti-rabbit Alexa Fluor Plus 647 (#A32733 at 1:1,000; Invitrogen/Molecular Probes) secondary antibodies. Slides were washed again in PBST and DNA was stained with Hoechst. Alternatively, a double immunofluorescence assay was performed by incubating simultaneously with a mix of anti-H3K4me2 and rabbit anti-H3 (#39163 at 1:500; Active Motif) primary antibodies. Washing steps and incubation with goat anti-rabbit Alexa Fluor Plus 647 were as explained above. Images were acquired with a YOKOGAWA CQ1 spinning disk confocal microscope. Maximum intensity projections of whole germlines acquired with the same settings were opened in ImageJ, and 10 germlines per strain over two biological replicates were processed identically. Intensities of antibody signal from the distal germline (mitotic and transition zones) or the mid-germline (early and mid-pachytene regions) were measured, normalized to either Hoechst or H3 signal, and averaged. Because comparable results were obtained regardless of the normalization used (Fig S2A and B), we routinely used Hoechst-based normalization.

## Immunoprecipitation for proteomics

Immunoprecipitations were performed on frozen embryos prepared by hypochlorite treatment from animals grown at 20°C on enriched NGM seeded with concentrated HB101 bacteria. After hypochlorite treatment, embryos were washed once in IP buffer (50 mM HEPES/KOH, pH 7.5; 300 mM KCl; 1 mM EDTA; 1 mM MgCl$_2$; 0.2% Igepal-CA630; and 10% glycerol) and flash-frozen in liquid nitrogen after the addition of beads. Embryos were then grounded to powder, resuspended in IP buffer containing complete protease inhibitors (Roche) and sonicated on ice at an amplitude of 30% for 2.5 min (15'' ON/15'' OFF pulses) using an Ultrasonic Processor (Bioblock Scientific). Protein extracts were recovered in the supernatant after centrifugation at 20,000$g$ for 15 min at 4°C. Protein concentration was estimated using the Bradford assay (Bio-Rad Protein Assay Dye).

For all GFP immunoprecipitations, 70 mg of total protein extract was incubated for preclearing with 200 $\mu$l slurry of binding control magnetic agarose beads (ChromoTek bmab-20) in IP buffer for 1 h at 4°C. Then 200 $\mu$l of GFP-TRAP MA beads slurry (ChromoTek) were added to each sample and incubation continued for additional 3 h on a rotator. Beads were collected with a magnet, washed three times in IP buffer and once in Benzo buffer (HEPES/KOH 50 mM, pH 7.5; KCl 150 mM; EDTA 1 mM; MgCl$_2$ 1 mM; Igepal-CA630 0.2%; and glycerol 10%). Beads were then incubated in 400 $\mu$l of Benzo buffer containing 2,500 units of benzonase (Sigma-Aldrich) for 1 h at 4°C and washed three times in IP buffer. Eluates were recovered by incubation at 95°C for 10 min in 60 $\mu$l of 1× LDS buffer (Thermo Fisher Scientific). 1/10 of each eluate was resolved on a 4–12% NuPage

Novex gel (Thermo Fisher Scientific) and stained with SilverQuest staining kit (Thermo Fisher Scientific). 40 µl of the eluates was then analyzed by mass spectrometry.

For HA::WDR-5.1 co-immunoprecipitation, samples were prepared as described previously (48).

### Mass spectrometry-based proteomic analyses

For the GFP co-immunoprecipitation: The eluted proteins were stacked at the top of a 4–12% NuPAGE gel (Invitrogen). After staining with R-250 Coomassie Blue (Bio-Rad Laboratories), proteins were digested in-gel using modified trypsin (sequencing purity; Promega), as previously described (78). The resulting peptides were analyzed by online nanoliquid chromatography coupled to MS/MS (Ultimate 3000 RSLCnano and Q-Exactive HF; Thermo Fisher Scientific) using a 120 min gradient. To this end, peptides were sampled on a precolumn (300 µm × 5 mm PepMap C18; Thermo Fisher Scientific) and separated in a 75 µm × 250 mm C18 column (Reprosil-Pur 120 C18-AQ, 1.9 µm; Dr. Maisch). The MS and MS/MS data were acquired by Xcalibur (Thermo Fisher Scientific).

Peptides and proteins were identified by Mascot (version 2.7.0; Matrix Science) through concomitant searches against the Uniprot database (*C. elegans* taxonomy, January 2021 version), classical contaminants database (homemade), and the corresponding reversed databases. Trypsin/P was used for digestion and two missed cleavages were allowed. Precursor and fragment mass error tolerances were set at 10 and 20 ppm, respectively. Peptide modifications allowed during the search were carbamidomethyl (C, fixed), acetyl (Protein N-term, variable), and oxidation (M, variable). The Proline software (79) was used for the compilation, grouping and filtering of the results (conservation of rank 1 peptides, peptide length ≥ 6 amino acids, peptide score ≥ 25, and false discovery rate of peptide-spectrum-match identifications <1% as calculated on peptide-spectrum-match scores by employing the reverse database strategy). Proline was then used to perform a compilation, grouping, and spectral counting-based comparison of the protein groups identified in the different samples. Proteins from the contaminant database were discarded from the final list of identified proteins. To be considered as a potential CFP-1 interactor, a protein must be identified with a minimum of three specific spectral counts and not in the negative control eluate, or enriched at least five times relative to the negative control eluate.

For the HA::WDR-5.1 mass spectrometry, samples were prepared and analyzed as described previously (48).

## Data Availability

The mass spectrometry proteomics data have been deposited to the ProteomeXchange Consortium via the PRIDE (PubMed ID: 30395289) partner repository with the dataset identifier PXD029904.

## Supplementary Information

## Acknowledgements

Strains were obtained from the Caenorhabditis Genetics Center (CGC), supported by the National Institutes of Health Office of Infrastructure Programs (P40 OD010440), and the National BioResource Project. We acknowledge the Discovery Platform and Informatics Group at EDyP (U1292 INSERM/CEA/UGA, Institut de Recherche Interdisciplinaire de Grenoble, IRIG), and the SFR Biosciences (UAR3444/CNRS, US8/Inserm, ENS de Lyon, UCBL): PLATIM. We thank Cécile Bedet, Laura Faccin, and Lisa-May Alvarez for technical support. We also thank Anne O'Rourke for critical reading of the manuscript and Florence Solari and Valérie Robert for helpful discussions. This project was supported by the Agence Nationale de la Recherche (ANR) [15-CE12-0018-01] and [ANR-19-CE12-0025] and the Centre National de la Recherche Scientifique (CNRS). The proteomic experiments were partially supported by ANR under projects ProFI (Proteomics French Infrastructure, ANR-10-INBS-08) and GRAL, a program from the Chemistry Biology Health (CBH) Graduate School of the University of Grenoble Alpes (ANR-17-EURE-0003).

### Author Contributions

M Caron: conceptualization, formal analysis, supervision, investigation, visualization, and writing—review and editing.
L Gely: formal analysis, investigation, and visualization.
S Garvis: formal analysis, investigation, and writing—editing.
A Adrait: formal analysis, data curation, and investigation.
Y Couté: formal analysis, data curation, and investigation.
F Palladino: conceptualization, funding acquisition, project administration, and writing—original draft, review, and editing.
P Fabrizio: conceptualization, project administration, validation, formal analysis, supervision, investigation, visualization, and writing—original draft, review, and editing.

### Conflict of Interest Statement

The authors declare that they have no conflict of interest.

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
