## [Reviewer comments · Life Science Alliance]

Life Science Alliance

Loss of SET1/COMPASS methyltransferase activity reduces lifespan and fertility in *Caenorhabditis elegans*

Matthieu Caron, Loic Gely, Steve Garvis, Annie Adrait, Yohann Couté, Francesca Palladino, and Paola Fabrizio

DOI: <https://doi.org/10.26508/lsa.202101140>

Corresponding author(s): Paola Fabrizio, École Normale Supérieure de Lyon and Francesca Palladino, École Normale Supérieure/Université de Lyon

Review Timeline:

Submission Date:	2021-06-22
Editorial Decision:	2021-07-30
Revision Received:	2021-10-28
Editorial Decision:	2021-11-15
Revision Received:	2021-11-25
Accepted:	2021-11-26

Scientific Editor: Novella Guidi

Transaction Report:

July 30, 2021

Re: Life Science Alliance manuscript #LSA-2021-01140

Dr. Paola Fabrizio
ENS de Lyon
46 allée d'Italie
Lyon 69007
France

Dear Dr. Fabrizio,

Thank you for submitting your manuscript entitled "Loss of SET1/COMPASS methyltransferase activity reduces lifespan and fertility in *C. elegans*." to Life Science Alliance. The manuscript was assessed by expert reviewers, whose comments are appended to this letter. As you will note from the reviewers' comments below, all the reviewers are quite positive and excited about the work that in their view presents several novel findings that are relevant to epigenetic regulation of aging. However, they do raise some concerns that would need to be addressed in the revised version before resubmission. In particular, Reviewer 2 and 3 have only few minor points that need to be addressed in the manuscript text, while Rev1 is more critical and raises different major concerns to be addressed. We, thus, encourage you to submit a revised version of the manuscript back to LSA that responds to all of the reviewers' points including the concern whether the mutation is really catalytically dead by testing in vitro the H1447K point mutation to ensure that catalytic activity is effectively lost. Also, because of the lack of internal control in the germline immunostaining, Reviewer 1 suggests to do co-IF by including a pan-histone 3 or a post-translational mark not related to H3K4 methylation to ensure that the reduction in the signal is not caused by a technical problem.

Thank you for this interesting contribution to Life Science Alliance. We are looking forward to receiving your revised manuscript.

Sincerely,

-- Summary blurb (enter in submission system): A short text summarizing in a single sentence the study (max. 200 characters including spaces). This text is used in conjunction with the titles of papers, hence should be informative and complementary to the title and running title. It should describe the context and significance of the findings for a general readership; it should be

written in the present tense and refer to the work in the third person. Author names should not be mentioned.

B. MANUSCRIPT ORGANIZATION AND FORMATTING:

Reviewer #1 (Comments to the Authors (Required)):

This is a study seeking to address whether depletion of H3K4me2/me3 positively or negatively contribute toward ageing in *C. elegans*. The authors are using a CRISPR/Cas9 generated strain that produces a single amino acid change in SET-2. By replacing a histidine at position 1447 by a lysine, the authors claim that the catalytic activity of SET-2 (H3K4 methylation) is dead. This mutation seems to have been designed based on previous work performed in yeast. They make a comparative analysis with a predicted null allele of *set-2*, bn129 as well as with a mild reduced function allele, ok952. Using the *set-2*(syb2085) allele they show that H3K4me2/me3 is reduced in the germline and embryos using IF and western blots. They also demonstrate that the SET-2H1447K can associate with the other components of the COMPASS complex. The contentious point that they try to resolve is a about how SET-2 influences lifespan. They use all three alleles and found that lifespan is reduced and not enhanced as another group has previously reported. This previous work has also linked this extended lifespan to accumulation of fat, but this phenotype was not detected in this present study in any of the *set-2* alleles studied.

The identification of a catalytic dead SET-2 is a significant finding and will be useful for other studies. It is also useful for the field to be aware of the discrepancies existing in regards of the role in H3K4 methylation and ageing.

Major points of concern

1. Is *set-2* (syb2085) really catalytically dead?

The syb2085 is presumed devoid of methylation activity, yet there is at least 20% of H3K4me2/me3 detected in germlines or embryos. Where would this ~20% of H3K4me2/me3 coming from? What about H3K4me1?

In addition, the result section state;" similar results were obtained when the germlines were immunostained for H3K4me2...". This is not corresponding to the presented data. The *set-2*(bn129) has far less H3K4me2 than *set-2*(syb2085), see figure 1C panels MZ+TZ H3K4me2 and PA H3K4me2. I feel using the term similar is not appropriate, since the comparison between these two alleles is used to promote the view that this allele is catalytically dead. I would rather re-phrase this part to better reflect the data.

I think the H1447K point mutation should be tested *in vitro* to ensure that catalytic activity is effectively lost. This has been previously performed for *set-16* (Fisher et al., *Dev Biol*, 2010)

It would also help to provide alignment between species and cite papers that have actually investigated this point mutation at a biochemical level. The paper cited by the authors basically state that it is a catalytic dead mutant, but I could not find evidence other than the incapacity for the presumed equivalent mutant to rescue a growth phenotype in yeast. At least I could not find any details in the paper cited (Rizzarda et al., *Genetics*, 2012) and another uncited paper (Schlichter and Cairns, *EMBO J*, 2005).

2. germline immunostaining lack internal control

Most studies are now using internal controls when performing immunostaining, since the signal of interest can be affected by a number of technical factors. I would suggest to perform co-IF by including a pan-histone 3 or a post-translational mark not related to H3K4 methylation. This would ensure that the reduction in the signal is not caused by a technical problem. I also understand that a low number of 5 germlines analysed. Were they from different biological replicates? This is surprising since worms are easy to grow in large number and that each slide can accommodate 10 gonads.

3. Western blot

What stage embryos have been used?

4. Lifespan part of the paper.

There is a big claim that this paper will resolve/explain contradictory results, but there are only one panel addressing *set-2*

regulated lifespan. The other group from which they reported an increase in lifespan used multiple strains and RNAi conditions over 3 separate studies. In addition, in light of the Lee et al. paper (Lee et al., *Elife*, 2019) would it not be more helpful to test whether later generations are displaying an extended lifespan? An explanation needs to be provided to explain the differences between these separate studies.

5. Oil Red O staining

Could the author add a mutant depleted in lipid droplet as well? Could we have a higher resolution and more zoomed in pictures? How many times was this experiment repeated?

6. I feel that the language used in the text is sometimes not reflecting the data, as explained above. But there is also a lack of precision when using 'global H3K4 methylation' and limiting the analysis to H3K4me2/me3 levels. For example, the previous studies are clearly stating that it is a reduction in H3K4me3 and not a loss of H3K4 methylation as written in several places in the manuscript. This is very different and important.

Overall, there is interesting data pointing towards a discrepancy with the published data on *set-2* and its role in longevity. However, it would only be beneficial to the field if the authors could show the condition in which alterations in SET-2 expression can extend lifespan and the condition in which lifespan is shortened. This has been achieved in the field for a number of mitochondrial mutants. Researchers have shown that a slight depletion of specific mitochondrial electron transport chain components increase lifespan and as the depletion becomes more robust lifespan is shortened to eventually reach lethality. In addition, in the case of WDR-5, the finding by Lee et al. that it takes several generations for the extended lifespan to manifest provides a satisfactory explanation with an apparent discrepancy. Finally, I think the concept that later *set-2* mutant generations may be long-lived is an avenue worthwhile exploring.

Reviewer #2 (Comments to the Authors (Required)):

Caron, Palladino and Fabrizio present an interesting analysis of the *C. elegans* methyltransferase SET-2, which is a large protein that is meant to methylate the transcriptional activation mark H3K4me3. SET-2 has been previously linked to longevity, as loss of SET-2 or the associated subunits WDR-5 or ASH-2 have been previously suggested to promote longevity. WDR-5 loss must occur for more than 10 generations to see adult longevity, and RNAi knockdown of ASH-2 or WDR-5 apparently also causes longevity. Loss of *set-2* has been previously shown to cause sterility following growth at the restrictive temperature of 25 degrees celsius. More recently, an independent group (Katz) has shown that deficiency for *wdr-5* for many generations results in longevity and that this longevity may depend on a rise in H3K9 methylation created by MET-2. Loss of WDR-5 and MET-2 causes short life. Furthermore, loss of the JMJD-1 H3K9 demethylase, which may mimic high levels of H3K9me that occur when WDR-5/SET-2/ASH-2 activity is suppressed, causes longevity. Together, these data suggest that loss of WDR-5 H3K4 methylase activity for multiple generations can lead to low H3K4me and high H3K9me, accompanied by longevity.

The current manuscript studies the potential role of the large SET-2 methyltransferase protein in regulation of fertility and longevity by generating a point mutation in the C-terminal methyltransferase domain that is predicted to inactivate all methylation activity. The authors do a nice job of demonstrating that catalytically inactive SET-2 leads to a global reduction in H3K4 methylation, in contrast to the *ok952* deletion allele of *set-2*, which removes a substantial portion of SET-2 but leaves the catalytic domain intact and may only modestly alter H3K4 methylation levels in germline. In contrast, two strong alleles of *set-2*, *by129* and the novel *syb2085* mutation, cause substantial 60-70% reductions in H3K4 methylation. The authors demonstrate that both strong alleles of *set-2*, but not the *ok952* deletion, cause previously reported defects for *set-2* that include reduced fertility and complete sterility after growth for several generations. This links the methyltransferase activity of SET-2 and potentially levels of H3K4 methylation with maintenance of fertility at high temperature. On the other hand, this work raises several questions about how SET-2 regulates longevity and why positive and negative effects have been reported by distinct groups. Discordant results discussed here raise an important topic that is not commonly addressed in the literature, but this topic is highly relevant to the scientific community.

On the other hand, the authors offer several surprising observations about longevity. All three alleles of *set-2* cause short lifespan, as does a loss of function allele of *cfp-1*, another subunit of the SET-2 methyltransferase complex. The lifespan experiments are done rigorously, in several independent trials with high n values. The lifespan assays are done in a similar manner to the assays of Greer et al, which originally reported long life for mutation of *set-2* and for RNAi of *ash-2*, at 20 degrees celsius without the sterilizing agent FUDR. However, *set-2 syb2085* was maintained as a balanced heterozygote until lifespan assays were initiated, which may have precluded accumulation of histone modifications in various segments of the genome that might have been present due to propagation of mutants for many generations in the original Greer paper. Overall, the authors have strong evidence for an effect of *set-2* mutation on lifespan, independent of generations propagated and independent of histone methyltransferase activity.

In general, this is an excellent paper that presents several novel findings that are relevant to epigenetic regulation of aging and fertility.

Comments:

1. In the Discussion, the authors suggest 'we found that set-2-mediated H3K4 methylation is essential for normal lifespan'. I was uncertain about this conclusion, as the authors demonstrate little or no effect of ok952 on H3K4 methylation levels in embryonic extracts and only a modest effect in germlines.
2. The authors present nice oil red O staining to show a second physiological effect that is not present in their research group. Together, their results indicate that set-2 ok932 behaves differently under conditions that the authors use in comparison to conditions in the Brunet group at Stanford. Perhaps contact Brunet and Katz about media ingredients and protocols, to see if there might be one or several simple explanations regarding discordant measures (ie, different sources of agar).
3. The authors present nice mass spectrometry data on proteins associated with the CFP-1 subunit in set-2 mutant backgrounds. I was uncertain about the significance of the different numbers reported in the Mass Spec figures. Why is DPY-30 so low? Are the higher levels of SET-2 for syb2085 reflective of negative feedback of SET-2 activity on SET-2 levels. Were there any other proteins whose levels varied in a manner that is specific to both syb2085 and ok932 such that their levels might explain the common phenotype of short life?
4. Perhaps the short life is due to loss of normal DAF-16 activity, which promotes longevity in wildtype and likely requires transcriptional activation marks such as H3K4me.

Minor:

Fig. 2c: the legend says blue and gray, but I see light gray and dark gray?

Fig. 4b: syb285 should be syb2085.

Please put the supplemental CFP-1 data in the main figures, as it offers a nice control.

Reviewer #3 (Comments to the Authors (Required)):

In this work Caron et al investigate the impact of H3K4 methylation on lifespan in *C. elegans* by creating a precise catalytic mutant. Loss of H3K4me reduces lifespan in most organisms but paradoxically has been reported to increase lifespan in worms. However, the authors find that actually the catalytic mutant decreases lifespan in line with other species. Furthermore they convincingly fail to reproduce the lifespan extension previously reported for a hypomorphic mutant ok952.

This work has been well performed and makes a useful contribution to the debate. Experiments are mostly of high quality and statistics are good. I think it could be published as is but a few minor changes would be helpful.

The really critical results here are in Figure 3 as they directly contradict published high impact results from another group. That is ok - I have little doubt that if the authors have struggled to repeat those results then others will too. However, reading the M&Ms I am not clear where this ok952 mutant came from. I guess from a stock centre, but in which case it is not clear that what the authors are using is actually the same strain as the other group used. Please make this absolutely specific. The simplest explanation for the difference in results is that the strain from the stock centre or from the other lab is either not ok952 or has acquired a second site mutation. This could also arise if the line used for outcrossing (N2) is different - it is not hard to imagine that the original N2 used across the world has diverged between labs. One obvious question is whether the authors have the ok952 line from the other lab and does that have the same lifespan effect?

Minor points:

What does the * in the ok952 indicate in Fig. 1A?

Throughout: SEMs are not terribly helpful as they do not convey spread of the data. I recommend the authors show SD, or (even better) individual better points with a line indicating the mean. This is up to them though.

Fig. 2C - What is the 'control'? Also the blue is gray in my pdf

Table S2 - I am not sure what is going on here... the positive sample is marked as previously published (by the same lab) but the mutant not. It seems important that if the positive and the mutant are to be compared they should come from the same

experiment. Otherwise the loss of the SET2 complex could be attributed to the IP being worse for technical reasons (particularly as the counts for the SET16 and core complexes are down including the WDR5 component that was IP'd). I think what is meant is that all three samples shown here were performed as one experiment but only the set-2 was previously published, which would be fine but should be made clear. If that is not the case I'm not sure this data would be presentable - but I don't think it is critical to the paper either.

In the discussion sentence "overexpression of the RBR-2 H3K3me3 demethylase" I assume they mean H3K4.

Response to Reviewers

Reviewer 1

We thank Reviewer 1 for her/his useful comments. Please find point by point responses to the concerns raised below.

1. *Is set-2 (syb2085) really catalytically dead?*

The syb2085 is presumed devoid of methylation activity, yet there is at least 20% of H3K4me2/me3 detected in germlines or embryos. Where would this ~20% of H3K4me2/me3 coming from? What about H3K4me1?

As discussed in previous publications from our lab and others, the residual H3K4me2/3 methylation is likely due to the activity of the only other SET1 family member in *C. elegans*, SET-16/MLL (Xiao et al PNAS 2011, doi: [10.1073/pnas.1019290108](https://doi.org/10.1073/pnas.1019290108); Li and Kelly PLoS Gen 2011, doi: [10.1371/journal.pgen.1001349](https://doi.org/10.1371/journal.pgen.1001349)). The lethality of *set-16* homozygous mutants has made analysis of its contribution to H3K4me2/3 challenging, although Fisher et al. 2010 (doi: [10.1016/j.ydbio.2010.02.023](https://doi.org/10.1016/j.ydbio.2010.02.023)) reported decreased H3K4me3 in *set-16/+* heterozygous mutants. In addition, we cannot rule out a redundant contribution of additional, less characterized SET domain proteins in *C. elegans* (Greer et al Cell Rep 2014 DOI: [10.1016/j.celrep.2014.02.044](https://doi.org/10.1016/j.celrep.2014.02.044); Engert et al, PLoS Gen 2018, DOI: [10.1371/journal.pgen.1007295](https://doi.org/10.1371/journal.pgen.1007295)). This information is now included in the Results section of the revised manuscript.

We did not specifically look at H3K4me1 because 1) this mark was unaffected following loss of SET-2 or WDR-5 in previous studies (Greer et al Nature 2011, DOI: [10.1038/nature09195](https://doi.org/10.1038/nature09195); Li and Kelly PLoS Genetics 2011; DOI: [10.1371/journal.pgen.1001349](https://doi.org/10.1371/journal.pgen.1001349) and our unpublished data), and 2) H3K4me1 deposition is dependent on MLL complexes in mammalian cells (Cheng et al Mol Cell 2014, DOI: [10.1016/j.molcel.2014.02.032](https://doi.org/10.1016/j.molcel.2014.02.032); Hu et al MCB 2013, DOI: [10.1128/MCB.01181-13](https://doi.org/10.1128/MCB.01181-13); Lee et al Elife 2013, DOI: [10.7554/eLife.01503](https://doi.org/10.7554/eLife.01503)).

In addition, the result section state;" similar results were obtained when the germlines were immunostained for H3K4me2...". This is not corresponding to the presented data. The set-2(bn129) has far less H3K4me2 than set-2(syb2085), see figure 1C panels MZ+TZ H3K4me2 and PA H3K4me2. I feel using the term similar is not appropriate, since the comparison between these two alleles is used to promote the view that this allele is catalytically dead. I would rather re-phrase this part to better reflect the data.

To assess whether the difference referred to is real, or instead reflects the inherent variability in IF experiments, we doubled the number of gonads analyzed for quantification (n= 10 gonads). This more complete analysis shows that the MZ+TZ and PA mean H3K4me2 signal in *set-2(bn129)* and *set-2(syb2085)* is less than 26% and 40% of wild-type, respectively (see revised fig. 1). Statistical analysis (one-way ANOVA) indicates that the difference between the mutants is not significant. Overall, we believe the use of the term "similar" is now justified.

I think the H1447K point mutation should be tested in vitro to ensure that catalytic activity is effectively lost. This has been previously performed for set-16 (Fisher et al., Dev Biol, 2010).

We agree that in theory an *in vitro* approach could be used to demonstrate loss of activity in SET-2(H1447K), as done for SET-16/MLL. However, it is now well established that additional COMPASS components (WRAD) are required for SET1/COMPASS H3K4me2/3 activity *in vitro* (as discussed above, H3K4me1 is catalyzed by MLL proteins, and is SET-2 independent in *C. elegans*; Shinsky et al JBC 2015, doi: [10.1074/jbc.M114.627646](https://doi.org/10.1074/jbc.M114.627646)). Previous attempts to detect H3K4 methyltransferase activity of recombinant yeast Set1 were unsuccessful (see Introduction in Williamson et al PLoS One 2013, DOI: [10.1371/journal.pone.0057974](https://doi.org/10.1371/journal.pone.0057974) and DOI: [10.1016/j.molcel.2005.07.024](https://doi.org/10.1016/j.molcel.2005.07.024) from the Shilatifard lab: “Recombinant Set1 by itself is not active in methylating histone H3”).

We have attempted production of a recombinant SET-2 protein in bacteria using different tags and strategies, but never succeeded in detecting the labeled protein. We suspect that tagged SET-2 proteins are unstable and rapidly degraded. We encountered similar difficulties when trying to label the endogenous protein using CRISPR-Cas9.

For all of the above reasons, the proposed strategy would require a substantial investment in time, co-expressing WRAD subunits together with SET-2 for HMT assays, preferably in a baculovirus system. As discussed below, based on thorough knowledge on the structure of SET domains and the experimental data we provide, the mutation we created is most likely catalytically dead.

It would also help to provide alignment between species and cite papers that have actually investigated this point mutation at a biochemical level. The paper cited by the authors basically state that it is a catalytic dead mutant, but I could not find evidence other than the incapacity for the presumed equivalent mutant to rescue a growth phenotype in yeast. At least I could not find any details in the paper cited (Rizzardi et al., Genetics, 2012) and another uncited paper (Schlichter and Cairns, EMBO J, 2005).

The reviewer is correct that in the previous version we did not provide sufficient information on the choice of the mutated residue. To address this point, we have now added an alignment of the catalytic SET domains from several HMTs (Fig. 1A), highlighting the highly conserved histidine residue (H1447 in *C. elegans*). Something that we should have more clearly stated is that structural studies and mutagenesis combined with *in vitro* assays have clearly demonstrated the importance of this conserved residue for the HMT activity of SET domain proteins (Wilson et al Cell 2002, DOI: [10.1016/s0092-8674\(02\)00964-9](https://doi.org/10.1016/s0092-8674(02)00964-9); Kwon et al EMBO 2003, DOI: [10.1093/emboj/cdg025](https://doi.org/10.1093/emboj/cdg025); Xiao et al Nature 2003, DOI: [10.1038/nature01378](https://doi.org/10.1038/nature01378); Cao et al PLoS One 2010, DOI: [10.1371/journal.pone.0014102](https://doi.org/10.1371/journal.pone.0014102); Trievel et al Cell 2002, DOI: [10.1016/s0092-8674\(02\)01000-0](https://doi.org/10.1016/s0092-8674(02)01000-0)). These important references have been added to the text along with a sentence that summarizes the evidence we relied upon in our choice of H1447K (see Results page 5).

As pointed out by the Reviewer, in the only paper we cited in the previous version, the importance of this conserved residue was inferred from the inability of the Set1(H1017K) mutation to rescue specific phenotypes associated with Set1 loss of function, i. e. growth under specific conditions (Rizzardi et al Genetics 2012, DOI: [10.1534/genetics.112.142349](https://doi.org/10.1534/genetics.112.142349)). However, in two subsequent articles that we now also cite (Soares et al.2014 and Williamson et al. 2013(DOI: [10.1016/j.celrep.2014.02.017](https://doi.org/10.1016/j.celrep.2014.02.017), DOI:

[10.1371/journal.pone.0057974](https://doi.org/10.1371/journal.pone.0057974)) the authors show complete loss of H3K4 methylation by WB analysis in yeast carrying the Set1(H1017K) and Set1(H1017L) mutant proteins.

In addition, a very recent paper (Abay-Norgaard, Development 2020, doi: [10.1242/dev.190637](https://doi.org/10.1242/dev.190637)) reported the construction of a *set-2* allele carrying the same mutation we generated. The authors observed a similar decrease in embryonic H3K4me3 for two *set-2* null alleles and the catalitically dead allele. Worms carrying the three alleles also show very similar neuronal defects.

Altogether, we believe that the evidence provided is consistent with *set-2(syb2085)* being catalytically inactive.

2. germline immunostaining lack internal control

Most studies are now using internal controls when performing immunostaining, since the signal of interest can be affected by a number of technical factors. I would suggest to perform co-IF by including a pan-histone 3 or a post-translational mark not related to H3K4 methylation. This would ensure that the reduction in the signal is not caused by a technical problem. I also understand that a low number of 5 germlines analysed. Were they from different biological replicates? This is surprising since worms are easy to grow in large number and that each slide can accommodate 10 gonads.

The issue of internal controls in IF analysis is highly relevant, but because each antibody has its own specificities, difficult to systematically address. We performed a test using anti-H3 antibodies as internal control and found that H3-based normalization did not differ substantially from Hoechst, validating our previous experiments. Nonetheless we were not entirely satisfied by the overall quality of H3 immunostaining and prefer to use Hoechst instead, as in many other recent studies (e. g. Yang et al PLoS Gen 2019, DOI: [10.1371/journal.pgen.1007992](https://doi.org/10.1371/journal.pgen.1007992); Kadekar and Roy PLoS biol 2019, DOI: [10.1371/journal.pbio.3000309](https://doi.org/10.1371/journal.pbio.3000309)). We are mentioning this control in our revised Methods (Fig. S2).

As mentioned above, we also doubled the number of gonads used for quantification of the IF signals to 10 (5 for each biological replicate). This information is now reported in the Methods.

3. Western blot

What stage embryos have been used?

Mixed staged embryos were used. Embryos were obtained by bleaching day-1 adults and immediately flash frozen. With this protocol we normally obtain mixed stage embryos (<100 cells). We added this information to the Methods.

4. Lifespan part of the paper.

There is a big claim that this paper will resolve/explain contradictory results, but there are only one panel addressing set-2 regulated lifespan. The other group from which they reported an increase in lifespan used multiple strains and RNAi conditions over 3 separate studies. In addition, in light of the Lee et al. paper (Lee et al., Elife, 2019) would it not be more helpful to test whether later generations are displaying an extended lifespan? An explanation needs to be provided to explain the differences between these separate studies.

Reviewer 1 may have overlooked the data shown in Table S3. Not only we used four different mutants for our lifespan studies: *set-2(ok952)*, *set-2(syb2085)*, *set-2(bn129)*, and *cfp-1(tm6369)*, but also tested each mutant in at least 3 independent trials. The Brunet Lab instead only used two mutants, *set-*

2(ok952) and *wdr-5(ok1417)*, and RNAi against *set-2*, *ash-2*, and *wdr-5*. We think our approach is more suited to assess the role of COMPASS dependent H3K4 methylation in longevity because 1) as mentioned in the Discussion, WDR-5 and ASH-2 are part of both SET-2/SET1 and SET-16/MLL complexes. Therefore, any phenotype observed in *wdr-5* or *ash-2* mutants could potentially be dependent, at least in part, on a loss of SET-16/MLL function; 2) WDR-5 also has COMPASS-independent functions (within the NSL complex, for example), which could contribute to any phenotype observed in *wdr-5* mutants 3) generally speaking, we prefer to use multiple alleles instead of RNAi, which often shows higher experimental variation and may have off-target effects.

The “transgenerational lifespan experiment” performed by Lee et al. is beyond the scope of our paper, and although we agree it could be informative, several months would be required to have enough replicate experiments. Our experimental set up was aimed at establishing how loss of COMPASS-dependent H3K4 methylation affects lifespan, without the added influence of confounding effects that may arise in mutant strains over time. In particular, by performing our experiments using either freshly thawed worms (wild-type, *set-2(ok952)*, and *set-2(bn129)*, or homozygous mutants obtained from a balanced strain for *set-2(syb2085)*), we wanted to avoid the accumulation of secondary epigenetic effects that may potentially affect lifespan. In these conditions, loss of SET-2 HMT activity is deleterious and shortens the lifespan of each mutant tested, as previously observed in budding yeast. This is the message we want to convey at this point. Nonetheless we agree with Reviewer 1 that SET1/COMPASS could function through different mechanisms to oppositely regulate lifespan in freshly thawed animals and in animals maintained for numerous generations. This could be addressed in follow-up studies.

5. Oil Red O staining

Could the author add a mutant depleted in lipid droplet as well? Could we have a higher resolution and more zoomed in pictures? How many times was this experiment repeated?

We added the mutant *eat-2(ad1116)* as a negative control. The experiments were performed twice with similar results and we are now showing larger representative images.

6. I feel that the language used in the text is sometimes not reflecting the data, as explained above. But there is also a lack of precision when using 'global H3K4 methylation' and limiting the analysis to H3K4me2/me3 levels. For example, the previous studies are clearly stating that it is a reduction in H3K4me3 and not a loss of H3K4 methylation as written in several places in the manuscript. This is very different and important.

We are using the term “global” H3K4 methylation as opposed to “site-specific specific”, as is common practice. To be more precise we have replaced "methylation" with "di-and trimethylation" in relevant passages. We have not done so systematically throughout the text to avoid redundancy and to preserve the flow of the text. We are careful not to use "absence of H3K4 methylation", but we feel "loss" is correct.

Reviewer 2

We thank Reviewer 2 for her/his very positive comments.

1. In the Discussion, the authors suggest 'we found that set-2-mediated H3K4 methylation is essential for normal lifespan'. I was uncertain about this conclusion, as the authors demonstrate little or no effect of ok952 on H3K4 methylation levels in embryonic extracts and only a modest effect in germlines.

As we mention in the Discussion, the fact that *set-2(ok952)* and all other mutants have a similarly short lifespan suggests that either lifespan is controlled by specific changes in methylation patterns on a handful of genes, or that there is a threshold for global H3K4 methylation, below which lifespan is reduced (this threshold being relatively high since, as pointed out by the Reviewer, the effect observed in *set-2(ok952)* is relatively small). Either way we think that the sentence: 'we found that *set-2*-mediated H3K4 methylation is essential for normal lifespan' is correct.

2. The authors present nice oil red O staining to show a second physiological effect that is not present in their research group. Together, their results indicate that set-2 ok932 behaves differently under conditions that the authors use in comparison to conditions in the Brunet group at Stanford. Perhaps contact Brunet and Katz about media ingredients and protocols, to see if there might be one or several simple explanations regarding discordant measures (ie, different sources of agar).

We indeed contacted the Brunet Lab at Stanford and compared our lifespan protocols. Unfortunately, we could not find any major difference that could justify the different outcomes of our experiments. Although we cannot formally rule out that the sources of agar or other reagents contribute to the variations observed, we find it unlikely that these differences could have such opposite effects on *set-2(ok952)* lifespan. As explained in the Discussion, a plausible explanation for the divergent results is the presence of an additional mutation/s in the genetic background of the strain used at Stanford. Alternatively, several generations of growth at 20°C and the accumulation of additional epigenetic marks may be required in order to see lifespan extension, as observed by the Katz Lab.

3. The authors present nice mass spectrometry data on proteins associated with the CFP-1 subunit in set-2 mutant backgrounds. I was uncertain about the significance of the different numbers reported in the Mass Spec figures. Why is DPY-30 so low? Are the higher levels of SET-2 for syb2085 reflective of negative feedback of SET-2 activity on SET-2 levels. Were there any other proteins whose levels varied in a manner that is specific to both syb2085 and ok932 such that their levels might explain the common phenotype of short life?

As indicated in the corresponding legend, the different numbers reported in Figure 2C correspond to spectral counts (SCs), i.e. the number of MS spectra identifying peptides belonging to individual proteins in each samples. This semiquantitative information is classically used to compare the protein contents of different samples.

Several hypotheses can explain the limited number of SCs obtained for DPY-30 in these analyses, the most probable being 1) the small size of the protein (~13 kDa) which makes it more difficult to detect (as a reminder, the proteins were digested with trypsin and the resulting peptides analyzed by MS ; only peptides of a given size (6 to ~25 amino acids) can be detected, and not all of them are detectable by MS), and 2) a lower abundance of this protein in immunoprecipitates compared to other proteins. However, the reproducible identification of DPY-30 with high quality peptide-spectrum matches makes us very confident about its presence in the analyzed complexes. We can provide spectra if necessary.

The SC-based comparison of samples cannot be used as an accurate measurement of protein abundances in these samples. Therefore, our results cannot be interpreted as proof of differential abundance of SET-2 in the analyzed complex samples, especially since the SC ratio is inferior to 2 between these different samples.

In this single experiment, no major differences were observed in the SCs of proteins found in the *set-2* mutants and wildtype animals.

4. Perhaps the short life is due to loss of normal DAF-16 activity, which promotes longevity in wildtype and likely requires transcriptional activation marks such as H3K4me.

This is an interesting possibility that could be explored in a follow-up study.

Minor:

Fig. 2c: the legend says blue and gray, but I see light gray and dark gray?

We fixed this.

Fig. 4b: syb285 should be syb2085.

We corrected figure 4.

Please put the supplemental CFP-1 data in the main figures, as it offers a nice control.

We moved CFP-1 data to Fig. 1C of the main text.

Reviewer 3

We thank Reviewer 3 for her/his very positive comments.

The really critical results here are in Figure 3 as they directly contradict published high impact results from another group. That is ok - I have little doubt that if the authors have struggled to repeat those results then others will too. However, reading the M&Ms I am not clear where this ok952 mutant came from. I guess from a stock centre, but in which case it is not clear that what the authors are using is actually the same strain as the other group used. Please make this absolutely specific. The simplest explanation for the difference in results is that the strain from the stock centre or from the other lab is either not ok952 or has acquired a second site mutation. This could also arise if the line used for outcrossing (N2) is different - it is not hard to imagine that the original N2 used across the world has diverged between labs. One obvious question is whether the authors have the ok952 line from the other lab and does that have the same lifespan effect?

We apologize for the confusion. We neglected to mention that our *set-2(ok952)* mutant was obtained from CGC (this information has now been added to the Methods), as was the *set-2(ok952)* used by the Brunet Lab (Greer et al, Nature 2011). So it is plausible that a second hypothetical life-extending mutation might have been present in the original CGC *set-2(ok952)* mutant (and not removed by

outcrossing in the Brunet Lab). Alternatively, as pointed out by the Reviewer, the second mutation could have originated from the N2 used for outcrossing.

We have not tested the mutant from the Brunet lab because in our hands the *set-2(ok952)* mutant consistently behaves as all other *set-2* mutants (they are all short-lived), and for the above-mentioned reasons. Since our experimental protocol is designed to avoid the accumulation of additional epigenetic marks, we think we have enough evidence to conclude that in our experimental conditions loss of SET-2 activity is deleterious.

Minor points:

*What does the * in the ok952 indicate in Fig. 1A?*

The asterisk indicates a small insertion (12 bp) present in the ok952 allele. We added this detail to the revised legend.

Throughout: SEMs are not terribly helpful as they do not convey spread of the data. I recommend the authors show SD, or (even better) individual better points with a line indicating the mean. This is up to them though.

We agree with the Reviewer and we replaced SEM with SD.

Fig. 2C - What is the 'control'? Also the blue is gray in my pdf

Control indicates the wild-type strain N2, which does not contain the transgene used for IP. We now explain this in the legend of Fig. 2C of the revised manuscript. We also changed blue to gray.

Table S2 - I am not sure what is going on here... the positive sample is marked as previously published (by the same lab) but the mutant not. It seems important that if the positive and the mutant are to be compared they should come from the same experiment. Otherwise the loss of the SET2 complex could be attributed to the IP being worse for technical reasons (particularly as the counts for the SET16 and core complexes are down including the WDR5 component that was IP'd). I think what is meant is that all three samples shown here were performed as one experiment but only the set-2 was previously published, which would be fine but should be made clear. If that is not the case I'm not sure this data would be presentable - but I don't think it is critical to the paper either.

It is as suggested by the Reviewer. All samples were processed at the same time but only the set-2 (+) data were previously published. We clarified this point in the comment below table S2.

In the discussion sentence "overexpression of the RBR-2 H3K3me3 demethylase" I assume they mean H3K4.

Yes. We fixed this.

November 15, 2021

RE: Life Science Alliance Manuscript #LSA-2021-01140R

Dr. Paola Fabrizio
École Normale Supérieure de Lyon
46 allée d'Italie
Lyon 69007
France

Dear Dr. Fabrizio,

Thank you for submitting your revised manuscript entitled "Loss of SET1/COMPASS methyltransferase activity reduces lifespan and fertility in *C. elegans*". We would be happy to publish your paper in Life Science Alliance pending final revisions necessary to meet our formatting guidelines.

- please incorporate Reviewer 2' remaining suggestions regarding the text structure, and few additional points to add in discussion, main text and Materials and Methods sections
- please add the Twitter handle of your host institute/organization as well as your own or/and one of the authors in our system
- please note that titles in the system and manuscript file must match
- please add an Author Contributions section to your main manuscript text
- please add a conflict of interest statement to your main manuscript text
- please consult our manuscript preparation guidelines <https://www.life-science-alliance.org/manuscript-prep> and make sure your manuscript sections are in the correct order
- we encourage you to revise the figure legend for figure 3 such that the figure panels are introduced in an alphabetical order
- please add callouts for Figures 1B, 3C, and S2A-B to your main manuscript text
- please add a Data Availability section and include Mass spectroscopy data deposition

Figures check:

- fig. S2A: please indicate the scale bare measurement in the figure legend
- please indicate molecular weights alongside protein blots

A. FINAL FILES:

-- Summary blurb (enter in submission system): A short text summarizing in a single sentence the study (max. 200 characters including spaces). This text is used in conjunction with the titles of papers, hence should be informative and complementary to the title. It should describe the context and significance of the findings for a general readership; it should be written in the

present tense and refer to the work in the third person. Author names should not be mentioned.

B. MANUSCRIPT ORGANIZATION AND FORMATTING:

Sincerely,

Reviewer #1 (Comments to the Authors (Required)):

I am happy with the revisions. The authors have addressed all my concerns.

I feel the identification of a catalytic dead SET-2 is a significant finding and will be useful for other studies. It is also useful for the field to be aware of the discrepancies existing in regards of the role in H3K4 methylation and ageing.

Reviewer #2 (Comments to the Authors (Required)):

The paper on SET-2 by Caron, Palladino and Fabrizio provides convincing evidence that mutation of set-2 results in shortening of lifespan. The authors have responded cordially and well to all of the reviewer comments. I have a few additional suggestions for improvement of this manuscript, but I am happy to recommend publication based on the current data.

Comments:

1. The authors present a novel set-2 mutant that is catalytically dead and phenocopies the strong allele of set-2 bn129 for a role of set-2 in germline immortality. Perhaps show this data first, which helps to emphasize that ok952 is not a strong allele but that syb2085 is.
2. The authors present data that contrasts with some data previously published by other groups, for example mutation of wdr-5 which is in the same complex as set-2. To this reviewer, it is puzzling that other groups report normal longevity for early

generation wdr-5 mutants and longevity after growth of wdr-5 mutants for 20-40 generations. This contrasts with the results of the authors, which suggest that late generation cfp-1, set-2 bn129, set-2 ok952 are short-lived, whereas early generation (balanced) set-2 syb2805 is also short-lived. This contrast with previously published results is difficult to understand from reading the text, unless one looks at the Methods. I suggest that the authors clearly explain in the text that previous reports have suggested that RNAi of set-2/ash-2 and mutants that were homozygous for many generations were suggested to live long, and that more recently it was shown that outcrossing or balancing of one subunit of the SET-2 complex, wdr-5, resulted in wildtype lifespan in early generations and that this became longer after some 20 generations. Then, in the Results section, the authors could state that they tested set-2 mutant lifespan in several ways, using some cfp-1 and set-2 mutants that have been homozygous for many generations, and also by using their set-2 syb2085 allele that was balanced and then examined for longevity after being homozygous for only a few generations. In all cases, the authors observed short lifespan, which contrasts with two publications but agrees with some publications.

3. The authors also should consider a discussion of possible reasons for short life of set-2 in some labs and long life in other labs. One step that would be helpful would be if the authors list their media ingredients in the Materials and Methods. Please list the company from which cholesterol was purchased, please indicate if cholesterol is added before autoclaving (as in the white wormbook) or after autoclaving (as recommended in the online wormbook). Cholesterol creates important molecules that are relevant to metabolism and longevity and may be a variable that contributes to lab-specific differences in longevity. Please indicate if vented or unvented plates were used for *C. elegans* lifespan experiments. Vented plates dry more rapidly and can be used in a day or two, unvented plates remain soupy for weeks. If the authors discuss these facts in the Methods, then future investigators will have ideas to test in terms of variables that might contribute to changes in set-2 mutant longevity observed in different labs.

4. A related issue is that the authors mention the different longevity results in the Discussion, but they do not mention what variables might be relevant. It would be helpful for readers if such variables were briefly mentioned in the Discussion and then more carefully detailed in the Methods.

5. There are several papers about variability in longevity experiments in different labs, as studied by the Caenorhabditis Intervention Testing Program, which are worth mentioning in this context. The central conclusion being that if all ingredients are from the same suppliers, that similar longevity is obtained in different labs:

Lithgow, Driscoll and Phillips, Nature 548 387-388
<https://www.nature.com/articles/548387a>

Plummer, T.W., Harke, J., Lucanic, M. Chen, E., Bhaumik, D., Harinath, G., Coleman-Hulbert, A.L., Dumas, K.J., Onken, B., Johnson, E., Foulger, A.C., Guo, S., Crist, A.B., Presley, M.P., Xue, J., Sedore, C.A., Chamoli, M., Chang, C., Chen, M.K., Angeli, S., Royal, M., Willis, J.H., Edgar, D., Patel, S., Chao, E.A., Kamat, S., Hope, J., Ibanez-Ventoso, C., Kish, J.L., Guo, M., Phillips, P., Lithgow, G.J., Driscoll, M. 2017. Standardized Protocols from the Caenorhabditis Intervention Testing Program 2013-2016: Conditions and Assays used for Quantifying the Development, Fertility and Lifespan of Hermaphroditic Caenorhabditis Strains Nature Protocols Exchange doi:10.1038/protex.2016.086

Lucanic, M., Plummer, WT, Chen, E., Harke, J., Foulger JC, Onken, B, Coleman-Hulbert, AL, Dumas, KJ, Guo, S., Johnson, E., Bhaumik, D., Xue, J., Crist, AB, Presley, MP, Harinath, G., Sedore, CA, Chamoli, M., Kamat, S., Chen, MK, Angeli, S., Chang, C., Willis, JH, Edgar, D, Royal, MA, Chao, EA, Patel, S., Garrett, T., Ibanez-Ventoso, C., Hope, J., Kish, JL, Guo, M, Lithgow, GJ, Driscoll, M., Phillips, P.C., Lithgow, G. 2017. Impact of genetic background and experimental reproducibility on identifying chemical compounds with robust longevity effects Nature Communications, 8:14256 doi 10.1038/ncomms1426. PMID: 28220799

Reviewer #3 (Comments to the Authors (Required)):

ALL my issues with this manuscript with minor and have been satisfactorily answered.

Response to Reviewers

We thank Reviewers 1 and 3 for their positive comments. Hereafter, please find the response to Reviewer 2 who had some additional comments on how to improve our revised manuscript.

Reviewer #2 (Comments to the Authors (Required)):

The paper on SET-2 by Caron, Palladino and Fabrizio provides convincing evidence that mutation of set-2 results in shortening of lifespan. The authors have responded cordially and well to all of the reviewer comments. I have a few additional suggestions for improvement of this manuscript, but I am happy to recommend publication based on the current data.

Comments:

1. *The authors present a novel set-2 mutant that is catalytically dead and phenocopies the strong allele of set-2 bn129 for a role of set-2 in germline immortality. Perhaps show this data first, which helps to emphasize that ok952 is not a strong allele but that syb2085 is.*

In the Results section titled "Loss of SET-2-dependent H3K4 methylation is responsible for loss of fertility" (page 8), which we think is the one the Reviewer is referring to, we do in fact mention the catalytically dead mutant first and then set-2(ok952). More generally, we agree that the emphasis should be on the new set-2(syb2085) mutant and indeed we describe the effect on H3K4methylation in set-2(syb2085) and (bn129) first, and then in set-2(ok952) (first section of the results).

2. *The authors present data that contrasts with some data previously published by other groups, for example mutation of wdr-5 which is in the same complex as set-2. To this reviewer, it is puzzling that other groups report normal longevity for early generation wdr-5 mutants and longevity after growth of wdr-5 mutants for 20-40 generations. This contrasts with the results of the authors, which suggest that late generation cfp-1, set-2 bn129, set-2 ok952 are short-lived, whereas early generation (balanced) set-2 syb2805 is also short-lived. This contrast with previously published results is difficult to understand from reading the text, unless one looks at the Methods. I suggest that the authors clearly explain in the text that previous reports have suggested that RNAi of set-2/ash-2 and mutants that were homozygous for many generations were suggested to live long, and that more recently it was shown that outcrossing or balancing of one subunit of the SET-2 complex, wdr-5, resulted in wildtype lifespan in early generations and that this became longer after some 20 generations. Then, in the Results section, the authors could state that they tested set-2 mutant lifespan in several ways, using some cfp-1 and set-2 mutants that have been homozygous for many generations, and also by using their set-2 syb2085 allele that was balanced and then examined for longevity after being homozygous for only a few generations. In all cases, the authors observed short lifespan, which contrasts with two publications but agrees with some publications.*

We think there may be a misunderstanding. In fact it is important to note that the Brunet Lab never explicitly reported the number of generations required to observe longevity extension in either set-2(ok952) or wdr-5 mutants. Based on the recent results published by the Katz Lab (doi: [10.7554/eLife.48498](https://doi.org/10.7554/eLife.48498)), we can only assume that the wdr-5 mutant used by the Brunet lab was outcrossed and/or freshly thawed and maintained for 20-40 generations before lifespan assessment.

Indeed the Katz lab observed that thawing can be used for lifespan "resetting", although this phenomenon has not been confirmed by other labs. Regardless, the Katz and Brunet labs clearly disagree on the effect of *wdr-5* RNAi, which is shown to require several generations to extend lifespan by Katz, but works right away according to Brunet. In our study we used 1) freshly thawed (F3-F4) wild-type, *set-2(ok952)*, *set-2(bn129)*, *cfp-1* animals (which according to Katz have been "reset") and 2) *set-2(syb2085)* homozygous animals obtained from a balanced strain. In both experimental set ups, SET1/COMPASS activity is essential for normal lifespan. We also think it is important to emphasize that the Katz Lab only tested *wdr-5* mutants. WDR-5 clearly has additional COMPASS independent functions as part of several other chromatin-associated complexes, so that SET1/COMPASS-independent mechanisms accounting for the longevity observed in *wdr-5* mutants after many generations cannot be ruled out. Overall, we feel that the interpretation of previous results based on "number of generations" is not straightforward to interpret or reproduce. For all these reasons, we prefer to discuss and cite previous results (e. g Introduction, end of page 4; Discussion, page 10), while providing a detailed protocol to facilitate the task of reproducing our own lifespan results. We hope the reviewer finds this satisfying.

3. The authors also should consider a discussion of possible reasons for short life of *set-2* in some labs and long life in other labs. One step that would be helpful would be if the authors list their media ingredients in the Materials and Methods. Please list the company from which cholesterol was purchased, please indicate if cholesterol is added before autoclaving (as in the white wormbook) or after autoclaving (as recommended in the online wormbook). Cholesterol creates important molecules that are relevant to metabolism and longevity and may be a variable that contributes to lab-specific differences in longevity. Please indicate if vented or unvented plates were used for *C. elegans* lifespan experiments. Vented plates dry more rapidly and can be used in a day or two, unvented plates remain soupy for weeks. If the authors discuss these facts in the Methods, then future investigators will have ideas to test in terms of variables that might contribute to changes in *set-2* mutant longevity observed in different labs.

We followed the suggestion of the Reviewer and we added a list of the reagents used to prepare worm media (page 13). We also indicated from which company individual reagents were purchased and provided additional details to help investigators with reproducibility.

4. A related issue is that the authors mention the different longevity results in the Discussion, but they do not mention what variables might be relevant. It would be helpful for readers if such variables were briefly mentioned in the Discussion and then more carefully detailed in the Methods.

Although we discussed several aspects that may explain the different longevity results, e. g. additional mutations present in the background, we did not mention other specific sources of lifespan variation, such as the purity of reagents, which are known to affect longevity. As suggested by the Reviewer, we added a sentence on this subject in the Discussion (page 10).

5. There are several papers about variability in longevity experiments in different labs, as studied by the Caenorhabditis Intervention Testing Program, which are worth mentioning in this context. The central conclusion being that if all ingredients are from the same suppliers, that similar longevity is obtained in different labs:

Lithgow, Driscoll and Phillips, Nature 548 387-388
<https://www.nature.com/articles/548387a>

Plummer, T.W., Harke, J., Lucanic, M. Chen, E., Bhaumik, D., Harinath, G., Coleman-Hulbert, A.L., Dumas, K.J., Onken, B., Johnson, E., Foulger, A.C., Guo, S., Crist, A.B., Presley, M.P., Xue, J., Sedore, C.A., Chamoli, M., Chang, C., Chen, M.K., Angeli, S., Royal, M., Willis, J.H., Edgar, D., Patel, S., Chao, E.A., Kamat, S., Hope, J., Ibanez-Ventoso, C., Kish, J.L., Guo, M., Phillips, P., Lithgow, G.J., Driscoll, M. 2017. Standardized Protocols from the Caenorhabditis Intervention Testing Program 2013-2016: Conditions and Assays used for Quantifying the Development, Fertility and Lifespan of Hermaphroditic Caenorhabditis Strains Nature Protocols Exchangedoi:10.1038/protex.2016.086

Lucanic, M., Plummer, WT, Chen, E., Harke, J., Foulger JC, Onken, B, Coleman-Hulbert, AL, Dumas, KJ, Guo, S., Johnson, E., Bhaumik, D., Xue, J., Crist, AB, Presley, MP, Harinath, G., Sedore, CA, Chamoli, M., Kamat, S., Chen, MK, Angeli, S., Chang, C., Willis, JH, Edgar, D, Royal, MA, Chao, EA, Patel, S., Garrett, T., Ibanez-Ventoso, C., Hope, J., Kish, JL, Guo, M, Lithgow, GJ, Driscoll, M., Phillips, P.C., Lithgow, G. 2017. Impact of genetic background and experimental reproducibility on identifying chemical compounds with robust longevity effects Nature Communications, 8:14256 doi 10.1038/ncomms1426. PMID: 28220799

We are now citing these important papers in our revised manuscript.

November 26, 2021

RE: Life Science Alliance Manuscript #LSA-2021-01140RR

Dr. Paola Fabrizio
École Normale Supérieure de Lyon
46 allée d'Italie
Lyon 69007
France

Dear Dr. Fabrizio,

Thank you for submitting your Research Article entitled "Loss of SET1/COMPASS methyltransferase activity reduces lifespan and fertility in *C. elegans*". It is a pleasure to let you know that your manuscript is now accepted for publication in Life Science Alliance. Congratulations on this interesting work.

DISTRIBUTION OF MATERIALS:

Again, congratulations on a very nice paper. I hope you found the review process to be constructive and are pleased with how the manuscript was handled editorially. We look forward to future exciting submissions from your lab.

Sincerely,
